# Content of Carnosic Acid, Carnosol, Rosmarinic Acid, and Proximate Composition in an Assortment of Dried Sage (*Salvia officinalis* L.)

**DOI:** 10.3390/molecules30234569

**Published:** 2025-11-27

**Authors:** Agnieszka M. Hrebień-Filisińska, Katarzyna Felisiak, Grzegorz Tokarczyk, Zuzanna Czachura, Kacper Kiliański

**Affiliations:** Department of Fish, Plant and Gastronomy Technology, Faculty of Food Sciences and Fisheries, West Pomeranian University of Technology in Szczecin, 70-310 Szczecin, Poland; katarzyna.felisiak@zut.edu.pl (K.F.);

**Keywords:** *Salvia officinalis*, bioactive components carnosic acid, rosmarinic acid, nutrient content

## Abstract

Due to the content of carnosic acid (CA), carnosol (C), and rosmarinic acid (RA), sage (*Salvia officinalis* L.) has antioxidant, anti-inflammatory, anticancer, antidiabetic, neuroprotective, antiaging, antimicrobial, and antiviral properties. However, current standardization and quality assessment procedures do not specify the content of these key substances in dried sage. The aim of this study was to analyze polyphenolic components, including CA, C, and RA, as well as basic nutrients, such as protein, fat, carbohydrates, and ash content in dried *Salvia officinalis*. Thirteen products available for sale in Poland were analyzed. HPLC studies have shown a very large variation in terms of CA, C, and RA (CA: 1.25–32.42 mg/g, C: 0–9.06 mg/g, and RA: 3.2–20.6 mg/g). Similarly, significant differences between samples were observed for the proximate composition. An appropriate system of standardization of herbs and spices could ensure the repeatability of the concentration of key, non-volatile active substances, which in adequate doses may have a beneficial effect on certain body functions, as well as on the shelf life and sensory characteristics of food.

## 1. Introduction

Sage (*Salvia officinalis* L.) is widely known for its versatile medicinal and culinary properties. It has antioxidant, anti-inflammatory, anticancer, antimicrobial, and antidiabetic effects, as well as the ability to improve mood and cognitive function [1,2,3]. In traditional medicine, it is used as a supportive agent for ailments and inflammations of the throat, mouth, and teeth, and as an expectorant for bronchial diseases. Sage also improves digestive function and may be helpful in preventing and treating symptoms of menopause, depression, obesity, dementia, and heart disease [4,5,6]. Moreover, it is considered an effective and natural food additive with preservative properties, as it counteracts microbiological activity and lipid oxidation [1]. This is due to its strong antioxidant potential, including the ability to scavenge free radicals [7] and its properties against various Gram-positive bacteria (*Streptococcus* sp., *Bacillus* sp., etc.), Gram-negative bacteria (*Escherichia coli*, *Klebsiella pneumonia*, *Pseudomonas* sp., etc.), and fungi [6]. Sage is also known as a culinary spice, primarily in Mediterranean cuisine, to enhance the flavor and aroma of meat dishes (veal, poultry liver, lamb, venison, ragout) and tomato sauces. It is characterized by a strong aromatic note and a pungent, bitter taste. Due to its health-promoting properties, it can also be a suitable component in the design of functional foods [1,8,9], which provide effects exceeding their basic nutritional values. Sage is therefore both a food product (spice) and a widely available herbal raw material. In addition to its traditional use in food and phytotherapy by household consumers, the plant is also used in the food, pharmaceutical, perfume, and cosmetics industries [10].

Sage is most often used dried or fresh (grown in pots or gardens) as a spice for dishes or as an herb for preparing infusions or herbal teas. It is also available in various herbal preparations, such as extracts, infusions, essential oils, and tablets. Using it in any form can contribute to, for example, improving health and food quality. However, the most popular and practical form is dried sage. The assortment of dried sage mainly includes sage leaves or upper shoots, cut or crushed. It is offered in various weights, most often loose or in bags/sachets, etc.

The wide use of sage is due to the presence of various bioactive compounds with diverse chemical compositions in its tissues. More than 75 active compounds have been identified in sage [11], which can be classified into various groups of chemical compounds, such as phenolic compounds, terpenes, alkaloids, and others [1,12]. Some of them, mainly terpenes, such as α- and β-thujone, camphor and 1,8-cineole, are volatile organic compounds and components of essential oils [12]. Sage is considered an oil-bearing plant, and the leaves and leafy shoot tops are a pharmacopoeial raw material [5]. The phenolic compounds of sage are mainly represented by phenolic acids (including rosmarinic acid, caffeic acid), flavonoids (including apigenin, luteolin, quercetin) and phenolic diterpenes (i.e., carnosic acid and carnosol) [1,13]. Among the various polyphenolic components of sage, carnosic acid and its derivative carnosol, as well as rosmarinic acid, deserve special attention, as they are among the most important compounds of sage, determining its wide range of benefits [13,14,15,16,17,18].

Consumers who consider sage not only a food additive but also a remedy for various ailments want to be sure they are choosing a high-quality product. However, sage quality can vary significantly, and a substandard product may not meet their expectations. The content of key active compounds in sage can fluctuate significantly, depending on various factors, such as cultivation, climate, and geographic location [3]. Additionally, the current standardization and quality assessment procedures do not guarantee the presence of key active substances in sage at appropriate levels. As a spice, sage is subject to general food law and food safety requirements. However, as a raw material with medicinal properties, it is subject to quality control by relevant authorities in the European Union and is standardized according to established guidelines (including the European Medicines Agency (EMA); the European Pharmacopoeia), according to which only tests of the essential oil are the basis for assessing its quality [19,20]. However, the content of polyphenolic compounds, which are no less important for its therapeutic effect, is completely omitted [21]. Therefore, commercially available sage may vary in levels of polyphenolic components. Scientific research on sage is also largely limited to examining the components of the essential oil [5,6,10,22]. However, there is little data on the content of key polyphenolic compounds in commercially available dried sage (*Salvia officinalis*), especially carnosic acid, carnosol, and rosmarinic acid, which are responsible for its strong antioxidant properties and comprehensive health-promoting effects [15,16,17,18]. There are also few publications presenting the results of sage’s basic nutritional components, which is crucial when using it as a functional additive in various food products. Enriching food with sage phytochemicals can model not only the content of bioactive components but also the composition of basic nutrients. Therefore, we decided to examine the content of key non-volatile antioxidant compounds and the basic chemical composition of dried sage available on the Polish market.

The aim of this study was to analyze the polyphenolic components, including carnosic acid (CA), carnosol (C), and rosmarinic acid (RA), as well as the essential nutrients, such as protein, fat, carbohydrates, and ash, in various commercially available dried sage (*Salvia officinalis*) products.

## 2. Results and Discussion

### 2.1. Bioactive Compounds

The tested assortment of dried sage (*Salvia officinalis* L.) consisted of loose-packed products (L), intended mainly for use as spices for dishes or for preparing extracts, as well as those packed in bags (B), i.e., typical herbal teas, with sage content in one bag ranging from 1.2 to 1.5 g. Most of the products were sage samples from conventional crops (C), and four sages came from organic farming.

The carnosic acid (CA) content in individual sage samples varied greatly, ranging from 1.25 to 32.42 mg/g of sage (Table 1). The highest carnosic acid content was determined in sage 1LC (32.42 mg/g), followed by 6LE (23.6 mg/g), and the lowest in 12BC (1.25 mg/g). Only four of the thirteen companies offered sage with carnosic acid content above the average, i.e., above 10.1 mg/g. Most of the sage samples tested were characterized by relatively low amounts of this component, ranging from 1.25 to 9.72 mg/g. However, the obtained CA contents in sage do not differ from the literature data [13,23,24]. In the study conducted by Abreu et al. [23], the content of CA in sage was 14.6 mg/g d.m., while in other species of the *Salvia* genus, it ranged from 0.1 to 20.1 mg/g d.m. Slightly lower values, ranging from 2.99 to 7.16 mg/g, were obtained in sage teas in the study conducted by Baskan et al. [13]. In our study, only one sage tea in a bag, i.e., 5BC, had a higher CA level (15.99 mg/g) than the average content; most sage samples in bags (B), were characterized by a lower concentration of this compound than in loose products (from 1.25 to 4.67 mg/g). In another study [24], depending on the extraction conditions with supercritical CO_2_, the content of CA in sage ranged from 0.29 to 120.0 mg/g.

Analogous trends and similar variability were observed for carnosol (Table 1). Its content ranged from 0 to 9.06 mg/g. In three of the thirteen sage products tested, no carnosol was identified at all. These products were also characterized by lower, below-average CA content. This may be due to the fact that carnosol is a derivative of carnosic acid, which is formed, among others, as a result of its oxidation [25]. Like carnosic acid, it is an unstable compound that can be transformed into other derivatives. Therefore, it can be assumed that its absence in the three products may result from both the low content of carnosic acid and its transformation into other derivatives.

In the study by Abreu et al. [23], the carnosol (C) concentration in *Salvia officinalis* was at the level of 0.4 mg/g d.m. However, a large range of C contents (from 4.1 to 15.1 mg/g d.m.) was obtained in 19 different specimens of *Salvia officinalis* in another study [26]. Comparing sage specimens the richest in C, the latter study yielded approximately 1.7 times higher C levels than the present study. This is most likely due, among other things, to the extraction method used. Lamien-Meda et al. [26] had used 50% methanol to isolate C from sage tissue, which could have resulted in a more intensive breakdown of CA to C compared to the concentrated methanol used in the present study. In our previous work, we also observed that carnosol, a derivative of carnosic acid, was more abundant in methanol–water extracts (70%) than in methanol extracts from sage macerates. According to Mulinacci et al. [27], CA is very sensitive to the presence of water during extraction, which may favor the conversion of carnosic acid to carnosol.

*Salvia officinalis* from various regions of Tunisia, depending on the habitat, has been found to contain from 3.278 mg/g to 6.000 mg/g dry plant material weight CA and from 5.045to 5.947 mg/g dry plant material weight C [28]. In contrast, fresh *Salvia officinalis* cultivated in Greece (Piperia Aridea region) has been found to contain from 3.001 to 7.033 mg CA and from 0.472 to 1.085 mg C per g fresh weight [29].

Rosmarinic acid (RA) was present in all sages. The average rosmarinic acid content in the tested samples was 15.1 mg/g, with significant variations in concentration between sages (Table 1). The highest rosmarinic acid content was found in sage 3LC (23.9 mg/g), followed by sage 1LC and 2LC (19.6–20.6 mg/g), while the lowest content was found in sage 12BC and 9LC (3.2–3.8 mg/g). The rosmarinic acid concentration in sage 12BC and 9LC was approximately 5–6 times lower than in sage 1LC.

In the study by Lamien-Meda et al. [26], the rosmarinic acid content in various sage specimens from gene banks also varied significantly, ranging from 5 to 47 mg/g d.m. According to Dragowić-Uzelac et al. [30], 1 g of dry sage, depending on the analysis conditions, can contain from 13.4 to 35 mg of RA. However, similar results to the sage varieties richest in this compound: 3LC, 1LC, and 2LC (19.6–23.9 mg/g) were also obtained in *Salvia officinalis* (22.87 mg/g) in our previous study [31], where the extraction was carried out in an identical manner (70% methanol, in a water bath at 85 °C for 30 min). Also, a somewhat similar RA content, ranging from 13.68 to 18.378 mg/g of dry plant matter, depending on the habitat in Tunisia, was obtained by Farhat et al. [28]. In another study [32] assessing the biological activity of various wild sage species from the Mediterranean region, *Salvia officinalis* was found to contain approximately 38.8 mg of RA per g of herbal material. Lower RA contents, ranging from 2.5 to 4.01 mg/g, were determined in sage teas by Baskan et al. [13]. The results obtained in this study are therefore similar to other published data [26,28,31], and differences between sage samples may be due to different extraction methods used for analysis, varying sage quality, and origin, as in the case of CA and C. Among the products with the highest RA content, only bulk-packaged products are found, while the sage with the lowest concentration of this ingredient is found in both: sachets and bulk.

Analyzing the sum of CA, C, and RA (Figure 1), the most important non-volatile bioactive compounds in sage, the highest concentrations were found in sage 1LC (57.9 mg/g), followed by 6LE (49.4 mg/g). The 1LC sample also had the highest CA content and a relatively high RA content (the chromatogram of sage 1LC is shown in Figure 2). The lowest concentrations were found in teas packaged in tea bags: 12BC and 4BC (4.8–7.1 mg/g). This may indicate that herbal product manufacturers may use lower-quality raw materials to produce tea bags. These tea bags may be composed, for example, of older plant shoots and contain a higher proportion of stem elements. Bioactive components are found primarily in the leaves and are almost absent in the lignified parts. It has been shown that young, upper rosemary leaves contain higher levels of carnosic acid and carnosol than older, lower leaves [33,34]. However, organic sage was not found to differ significantly from conventionally grown sage. Both organic and conventional products exhibited a wide range of concentrations of CA, C, and RA, encompassing both high and very low levels. However, within both bagged and bulk products, the significant variation in CA, C, and RA levels, apart from the analysis conditions, may generally result from the varying quality of the plants used to produce the products studied. According to Cuvelier et al. [35], the content of active substances in a plant depends primarily on the quality of the original plant, its geographical origin, climatic conditions, harvest date, storage method, etc.

In the studies of Farhat et al. [36], depending on the place of collection, sage contained from 13.680 to 18.378 mg/g d.m. rosmarinic acid, from 3.278 to 6.00 mg/g d.m. carnosic acid, and from 5.045 to 5.947 mg/g d.m. carnosol. Differences in the content of carnosic acid, carnosol, and rosmarinic acid depending on the geographical region of Turkey were also confirmed in another study [37]. Producers of the tested sage samples declared their country of origin only in 5 of the 13 cases. It was noticed (Figure 1) that the sage samples from Poland (7LC, 8LE, 10BE, 13LE) are characterized by a lower content of CA and its derivatives than the salvias from Bosnia and Herzegovina (6LE), but in terms of RA content, they are comparable to it (10BE, 13LE) and even exceed it (in the case of sample 8LE). Similarly, Yesil-Celiktas et al. [37] confirmed differences in the content of CA, C, and RA depending on the geographical region of Turkey where the plants were grown. The highest CA content was found in rosemary from the hottest region, while the highest RA content was found in plants from the coolest and hottest regions. According to the information on the packaging, four of the sage samples came from Poland, and one was from Bosnia and Herzegovina. The remaining sage samples could therefore have originated from different regions of the word, with different climatic conditions. However, they most likely originated in Poland, the Balkans, or the Mediterranean, as these are among the most important producers of the herb for Europe. This diversity in origin among the dried sage products could be a major factor contributing to the wide variation in concentrations of key active compounds.

EU regulations introduce the requirement to indicate the origin of food products in the case of organic products [38,39]; therefore, apart from organic products, most products did not have such information on the packaging.

Moreover, soil conditions and the content of mineral components in the soil [40,41], as well as irrigation [42], sunlight [43], season, and date of harvest [36,44,45] have a significant impact on the accumulation of active substances in plants during growth.

High variability in the content of rosmarinic acid, carnosol, and carnosic acid may also occur within the same *Salvia officinalis* specimens, growing in identical environmental conditions and tested in the same periods [26], which in turn results from the genotype of a specific plant.

Similarly, significant variations were observed in the content of total polyphenols and flavonoids in dried sage. The average polyphenol content in the sage samples tested, sourced from different companies, was 77.9 mg/g (Table 1). However, the concentration of these compounds was very uneven, ranging from 26.1 mg/g to 135.2 mg/g. The highest polyphenol content was found in sage 6LE (135.2 mg/g), followed by sage 1LC (124.3 mg/g) and 3LC (117.3 mg/g). The lowest amount of these compounds was found in sage 12BC (26.1 mg/g). Sage 12BChad more than five times lower polyphenol concentrations than the richest sage (6LE). Seven among thirteen tested sage samples (4BC, 7LC, 8LE, 9LC, 10BE, 11BC, 12BC) were characterized by low polyphenols content, below average (>77.9 mg/g). In the context of other studies, a similar polyphenol content in sage, ranging from 72.9 to 144.1 mg/g d.m., was obtained by Sadowska et al. [46]. They investigated the effect of drying on the components of herbs and determined total polyphenols in a very similar manner to that in the present study. Lower results of polyphenol concentration in sage (from 11.59 mg/g to 13.39 mg/g d.m.), depending on the drying method, despite comparable analysis conditions, were obtained by Kwaśniewska-Karolak and Mostowski [47], which may result not only from the influence of different drying methods, but also from different sources and varying plant quality. The variation in polyphenol content in sage depending on the drying method was also observed in another study [48].

In the study by Khiya et al. [49], conducted in Morocco, different levels of total polyphenols were obtained in sage depending on the region of origin (Boulemane region: 70.5 to 85.1 mg/g; Khenifra region: 170.5–176.5 mg/g of extract). Farhat et al. [36] obtained different polyphenol contents in sage (on average from 76.65 to 146.76 mg/g of dry plant matter), depending on the harvest date. The highest total polyphenol content was found in sage during the fruiting period, while lower amounts of these compounds were determined in the vegetative phase.

The flavonoid content in various sage batches ranged from 4.8 to 18.2 mg/g in the presented study, which is consistent with literature data [30]. Depending on the extraction method and solvent used, it can range from 2.3 to 16 mg/g [30]. Similar results for sage from Poland, ranging from 2.7 to 12.3 mg/g d.m., were also obtained by Sajewicz et al. [50]. The highest flavonoid content was found in sage 6LE (18.2 mg/g), followed by sage 1LC (16.8 mg/g), while the lowest was found in sage 12BC (4.8 mg/g). The distribution of flavonoids in the various sages was very similar to that of polyphenols. Flavonoids are a subgroup of polyphenols, therefore the flavonoid content in the plant influences the overall polyphenol content.

In summary, the tested dried sages available for sale in Poland varied significantly in terms of the concentration of non-volatile active compounds. This significant variation in CA, C, RA, total polyphenols, and flavonoids levels within the single species *Salvia officinalis* L. may be due to numerous factors. The content of active compounds in the plants depends on factors such as genetic characteristics, geographical origin, climatic conditions (humidity, sunlight, air temperature), agrotechnical conditions (fertilization, salinity, irrigation), harvesting period, type of organ (leaves or shoots), type of leaves (upper-young or lower-older), storage, drying conditions, etc. [3,28,33,35,36,37,41,45,48]. The higher concentration of active compounds in loose products than in sage teas packed in bags may be related to the fact that the teas may be produced from older plant shoots, containing parts of the stems, or even may be adulterated. Such a wide variation in commercially available sage may be concerning for consumers who seek a concentrated source of non-volatile active compounds with beneficial effects on health. According to general international standards [51], the quality of sage (*Salvia officinalis*) is determined, among other things, by the concentration of volatile oil in the dry matter of the sage, which should be at least 1.5% (mL/100 g) on a dry basis. The chemical composition of the oil is also taken into account [52]. Similarly, according to European Pharmacopoeia [20], the concentration of the oil (10–15 mL/kg of dried sage) is the main criterion determining its suitability as a medical product. Furthermore, the pharmacopoeia also includes guidelines for, among other things, thujone, foreign matter, water, and total ash. However, there are no standards specifying limits or appropriate content of other non-volatile active compounds of sage. Some of the most important biocompounds in sage include carnosic acid, carnosol, and rosmarinic acid, which have very beneficial effects on the body and improve food quality. However, these compounds must be present in adequate amounts in sage to be effective. Therefore, introducing an appropriate system for standardizing herbs and spices, or improving existing procedures, could reduce the degree of variability in sage quality and guarantee consumers consistent quality of generally available products.

### 2.2. Composition of Basic Nutrients and Moisture

The studied sage samples were characterized by varied proximate composition (Table 2).

Moisture content ranged from 6.58% to 13.22%, with no pattern related to sage tea form being observed, as each group contained leaves with both the highest and lowest moisture content. The smallest differences were observed only in sage from organic farming, approximately 12%. According to general international standards [51], the moisture content of sage should be a maximum of 12% (*w*/*w*). In the case of six sages in this study, these contents were slightly exceeded.

The protein content of the studied sage samples ranged from 7.36% to 16.20%, with the highest protein content found in conventionally grown sage sold loose. In the case of sage packaged in bags, the protein content was lower, only in one case reaching 14.69%, and in two cases it was below 10%.

Fat content in sage also varied—from 6.21% to 12.65%, with the lowest results recorded for conventionally grown sage sold in bags.

Total ash content varied statistically significantly, from 7.61% for organically grown sage packaged in bags to 13.41% for organically grown sage in bulk. On average, ash content ranged from 9 to 11%. Of the nine conventionally grown sage samples, six contained more than 10% ash, while one of the four organically grown samples contained more than 10%. According to the ISO 11165:1995 standard [51], the ash content in sage should be a maximum of 11% *w*/*w* on a dry basis. Pharmacopoeia [20] specifies an ash content of up to 10% for sage. In our study, only six sage samples had ash content below 10%.

Karagözoğlu and Kiran [53], who studied sage from different herbalists, also found differences in proximate composition; however, the fat content and ash content were generally lower than determined in the present study (3.32–5.09% and 8.11–6.39%, respectively). Vlaicu et al. [54] found the crude protein in sage of 9.56% which was similar to the concentration reported in the present study. However, they found lower content of crude fat (only 3.15%). Ash content was in the range of our results. In contrast, Tomescu et al. [55] found lipid and ash content similar to that in the present study, but crude protein content was lower—only 6.77%. Todorova et al. [56] however, found a significantly higher crude protein content—15.28% d.m., which was within the range of the results of this study.

In general, the proximate composition of herbs depends on many factors, including the location of cultivation, soil components, age, harvest time, plant part, storage, and drying conditions [53,57]. In the case of sage, the differences are sometimes statistically significant, but they do not affect the product’s stability or its characteristics. The most important factor determining sage quality is the presence of bioactive compounds. However, adding sage to food products can not only enrich them with polyphenols, among other things, but can also affect the content of basic nutrients.

## 3. Materials and Methods

### 3.1. Sage (Salvia officinalis L.)

Thirteen sage products from twelve different companies purchased in Poland were analyzed. Sage grown conventionally and from organic farms (certified organic farming) were examined. All products were dried and available in loose form and in tea bags (Table 3). All sage samples were ground to uniform particle size (particle diameter up to 0.4 mm). Analyses were performed in 3 independent replicates for each sample (n = 3).

### 3.2. Other Materials

Analytically pure chemical reagents (methanol, sodium carbonate, Folin–Ciocalteu reagent, aluminum chloride), HPLC solvents (glacial acetic acid, acetonitrile, water), and high purity (≥98%) standard substances (carnosol, carnosic acid, rosmarinic acid, from two manufacturers: Chempur (Piekary Śląskie, Poland) and Sigma-Aldrich (Darmstadt, Germany), were used in this study.

### 3.3. Preparation of Sage Extracts

Extracts were prepared to determine the active compounds in sage. All sage samples were extracted with methanol and a 70% (*w*/*w*) aqueous solution of methanol. In both cases, 0.05 g of ground sage and 50 cm^3^ of solvent were used for extraction. Extraction was carried out in a water bath under a reflux condenser at 85 °C for 30 min. The extracts were then cooled and filtered to obtain a clear liquid (they were not evaporated). Methanol extracts were used to determine carnosic acid and cranosol, while extracts with 70% methanol were used to determine rosmarinic acid, total polyphenols, and total flavonoids. Analyses were performed immediately after obtaining the extracts.

### 3.4. Determination of Total Polyphenol Content

The total polyphenol content (70%) in methanol–water extracts was determined spectrophotometrically using the Folin–Ciocalteu reagent according to the method of Singleton and Rossi [58]. The assay solution was prepared as follows: 1 cm^3^ of sage extract, 16 cm^3^ of distilled water, 2 cm^3^ of saturated sodium carbonate, and 1 cm^3^ of Folin–Ciocalteu reagent. The prepared solutions were then placed in the dark for 60 min. After this time, their absorbance was measured at a wavelength of λ = 760 nm. The results in mg/mL were calculated based on a standard curve (y = 0.16 x; R^2^ = 9987), which was prepared for various concentrations of gallic acid (from 0.1 to 1.5 mg/cm^3^). Polyphenols in the sage extracts were then converted to mg/g of sage.

### 3.5. Determination of Total Flavonoid Content

Flavonoids in methanol–water extracts (70%) were determined according to Chan et al. [59], with our own modification. Sage extract and a 2% solution of aluminum chloride in methanol were added to test tubes in a 1:1 volume ratio, and absorbance was read after 10 min at a wavelength of 420 nm. After this time, the prepared solution turned yellowish. The flavonoid concentration in 1 cm^3^ of extract was calculated from a standard curve (y = 0.0319 x; R^2^ = 9996), with a linear range from 0.01 to 0.1 mg/cm^3^ quercetin. The flavonoid content was then recalculated and finally presented in mg per 1 g of sage [mg/g].

### 3.6. Determination of Rosmarinic and Carnosic Acid and Carnosol Using Liquid Chromatography (HPLC)

The identification of the active compounds was carried out using liquid chromatography (HPLC). An Agilent 1260 Infinity II liquid chromatograph coupled with a PDA detector (Agilent Technologies, Santa Clara, CA, USA) was used. Separation was performed on a Nucleosil 120-5 C18 reversed-phase column, 250 × 4.6 mm, at room temperature. The mobile phase consisted of acetonitrile (solvent A) and water with 5% (*w*/*w*) acetic acid (solvent B). The flow was maintained at 0.5 mL/min. The gradient program was as follows: 15% A/85% B from 0 to 12 min, then changed linearly to 0% A/100% B after 30 min, then changed to 85% A and 15% B after 50 min and the 85% A and 15% B system was maintained for another 10 min. The total analysis time was 60 min. The injection volume was 20 μL, and peaks were monitored at λ = 280 nm (carnosic acid, carnosol) and 325 nm (rosmarinic acid). Extracts were filtered through a 0.45 μm membrane filter before injection. Carnosic acid and carnosol were determined in methanol extracts, and rosmarinic acid in methanol–water extracts. Compounds were identified based on spectrum and retention time; retention time [minutes]: carnosic acid: 24.4–24.5; carnosol—22.4–22.6; rosmarinic acid—11.9. The concentration of a given compound was determined based on previously prepared standard curves in mg per mL of extract and then converted and expressed as mg per g of sage.

### 3.7. Proximate Composition—Nutrients and Moisture

Total nitrogen, fat, moisture, and ash content were determined by AOAC methods, 978.04, 925.18, 925.19, 923.03, respectively [60]. Content of carbohydrates was calculated as a difference between one hundred percent and sum of the contents of remaining components.

### 3.8. Statistical Analysis

The results presented in this article represent the statistical mean of three replicates (*n* = 3), and the standard deviation (SD) was also calculated. Statistical analysis of data was performed using Statistica (version 12.3). A significance level of *p* < 0.05 was applied, and the analysis was performed using one-way analysis of variance (ANOVA) with Tukey’s post hoc test.

## 4. Conclusions

The studied sage samples available on the Polish market varied significantly in terms of their content of carnosic acid, carnosol, rosmarinic acid, as well as total polyphenols and flavonoids. Significant differences were also noted between the products tested in terms of their basic composition (protein, lipids, carbohydrates, ash, and moisture). Of the biocompounds analyzed, rosmarinic acid and carnosic acid were identified in all sage samples evaluated, while carnosol was absent in an average of one in fourth products. Comparing the average content of selected compounds in sage by cultivation type, no clear differences were found between organic and conventional sage. However, sage sold in tea bags was generally characterized by lower content of carnosic acid, carnosol, and rosmarinic acid, which may indicate a lower quality of the raw material than bulk-packed sage. An appropriate system for standardizing herbs and spices could ensure the repeatability of the concentration of key, non-volatile active substances, which, in appropriate doses, can have a beneficial effect on certain body functions, as well as on the shelf life and sensory characteristics of food.

## Figures and Tables

**Figure 1 molecules-30-04569-f001:**
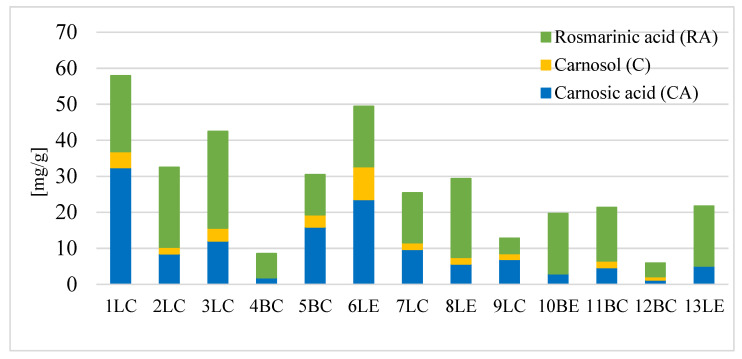
The sum of carnosic acid (CA), carnosol (C), and rosmarinic acid (RA) in the assortment of dried sage in mg/g of sage (1…13—sage code, L—loose sage, B—sage in bags, C—conventional cultivation, E—organic cultivation).

**Figure 2 molecules-30-04569-f002:**
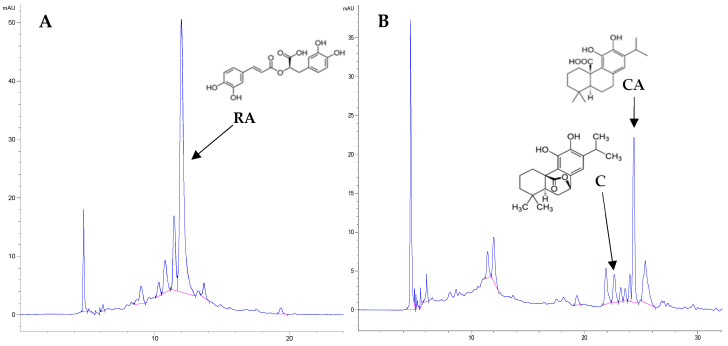
Chromatograms of the methanol–water extract (70%) (**A**) and the methanol extract (**B**) from *Salvia Officinalis* sample 1LC (RA—rosmarinic acid, retention time 11.95 min, λ = 325 nm; C—carnosol, retention time—22.66 min, λ = 280 nm; CA—carnosic acid, retention time—24.36 min, λ = 280 nm).

**Table 1 molecules-30-04569-t001:** Content of carnosic acid (CA), carnosol (C), rosmarinic acid (RA), and total polyphenol and flavonoid content in the assortment of dried sage in mg/g of sage (1…13—sage code, L—loose sage, B—sage in bags, C—conventional cultivation, E—organic cultivation).

Sage	CA [mg/g]	C [mg/g]	RA [mg/g]	Polyphenols [mg/g]	Flavonoids [mg/g]
1LC	32.42 ± 0.67 ^a^	4.45 ± 0.31 ^b^	19.6 ± 0.75 ^bc^	124.3 ± 2.5 ^b^	16.8 ± 0.5 ^a^
2LC	8.52 ± 0.15 ^k^	1.81 ± 0.06 ^d^	20.6 ± 0.60 ^b^	111.6 ± 6.6 ^c^	13.8 ± 0.4 ^c^
3LC	12.11 ± 2.10 ^d^	3.52 ± 0.17 ^c^	23.9 ± 2.34 ^a^	117.3 ± 6.5 ^bc^	16.1 ± 1.1 ^ab^
4BC	1.89 ± 0.05 ^l^	nd	6.2 ± 0.43 ^h^	58.7 ± 2.0 ^g^	7.1 ± 0.1 ^f^
5BC	15.99 ± 0.39 ^c^	3.34 ± 0.47 ^c^	9.9 ± 0.30 ^g^	92.0 ± 1.8 ^d^	12.3 ± 0.3 ^d^
6LE	23.60 ± 0.82 ^b^	9.06 ± 0.94 ^a^	14.9 ± 0.82 ^d^	135.2 ± 2.9 ^a^	18.2 ± 0.5 ^h^
7LC	9.72 ± 1.01 ^e^	1.85 ± 0.04 ^d^	12.1 ± 1.21 ^f^	48.1 ± 4.3 ^h^	13.9 ± 0.6 ^c^
8LE	5.70 ± 0.06 ^g^	1.85 ± 0.03 ^d^	19.2 ± 0.77 ^c^	64.4 ± 1.0 ^f^	12.9 ± 0.6 ^d^
9LC	7.00 ± 0.05 ^f^	1.58 0.08 ^e^	3.8 ± 0.32 ^i^	33.8 ± 1.3 ^i^	10.2 ± 0.9 ^ei^
10BE	2.98 ± 0.02 ^j^	nd	14.6 ± 1.52 ^d^	54.9 ± 6.2 ^gh^	11.4 ± 1.5 ^id^
11BC	4.67 ± 0.03 ^i^	1.85 ± 0.05 ^d^	13.0 ± 1.19 ^df^	49.9 ± 8.0 ^h^	15.3 ± 1.2 ^b^
12BC	1.25 ± 0.01 ^m^	0.95 ± 0.03 ^f^	3.2 ± 0.31 ^i^	26.1 ± 2.9 ^i^	4.8 ± 0.3 ^g^
13LE	5.12 ± 0.02 ^h^	nd	14.5 ± 0.64 ^d^	96.7 ± 6.0 ^d^	9.9 ± 0.6 ^e^
Average	10.1	2.2	15.1	77.9	12.5

^a–m^ values in the same columns, marked with the same lowercase letter, are not statistically significantly different, at *p* < 0.05; nd = not detected.

**Table 2 molecules-30-04569-t002:** Composition of basic nutrients and moisture content in dried sage products (1…13—sage code, L—loose sage, B—sage in bags, C—conventional cultivation, E—organic cultivation).

Sage	Moisture (%)	Protein (%)	Fat (%)	Ash (%)	Carbohydrates (%)
1LC	6.77 ± 0.34 ^c^	11.25 ± 0.75 ^c^	11.15 ± 0.40 ^bcd^	9.27 ± 0.27 ^c^	61.57 ± 1.42 ^b^
2LC	7.01 ± 0.29 ^c^	14.55 ± 0.47 ^b^	8.07 ± 0.57 ^ef^	13.13 ± 0.08 ^ab^	57.24 ± 0.46 ^c^
3LC	10.01 ± 0.47 ^b^	16.13 ± 0.10 ^a^	11.47 ± 0.66 ^ab^	10.48 ± 0.19 ^c^	51.92 ± 0.89 ^d^
4BC	6.58 ± 0.35 ^c^	10.63 ± 0.30 ^c^	6.63 ± 0.06 ^f^	9.08 ± 0.04 ^c^	67.07 ± 0.27 ^a^
5BC	9.82 ± 0.49 ^b^	11.94 ± 0.75 ^c^	9.81 ± 0.26 ^d^	9.19 ± 0.17 ^c^	59.25 ± 0.83 ^bc^
6LE	10.87 ± 0.11 ^b^	10.43 ± 0.18 ^c^	12.65 ± 0.76 ^a^	7.85 ± 0.36 ^d^	58.20 ± 1.02 ^c^
7LC	12.68 ± 0.45 ^a^	16.20 ± 0.50 ^a^	8.73 ± 0.40 ^de^	12.57 ± 0.09 ^ab^	49.82 ± 0.43
8LE	12.10 ± 0.55 ^a^	11.32 ± 0.21 ^c^	8.44 ± 0.50 ^de^	9.57 ± 0.55 ^c^	58.58 ± 1.56 ^c^
9LC	10.55 ± 0.61 ^b^	13.54 ± 0.34 ^b^	9.42 ± 0.07 ^de^	12.03 ± 0.46 ^b^	54.45 ± 1.05 ^d^
10BE	12.72 ± 0.45 ^a^	7.36 ± 0.27 ^e^	9.87 ± 0.37 ^de^	7.61 ± 0.02 ^d^	62.45 ± 0.67 ^b^
11BC	12.4 6 ± 0.53 ^a^	14.69 ± 0.74 ^ab^	7.57 ± 0.42 ^ef^	11.93 ± 0.21 ^b^	53.36 ± 1.38 ^d^
12BC	13.22 ± 0.28 ^a^	8.90 ± 0.67 ^d^	6.21 ± 0.84 ^f^	11.84 ± 0.09 ^b^	59.84 ± 0.94 ^bc^
13LE	12.32 ± 0.13 ^a^	10.93 ± 1.12 ^c^	8.98 ± 0.40 ^de^	13.41 ± 1.25 ^a^	54.36 ± 2.44 ^d^

^a–f^ values in the same columns, marked with the same lowercase letter, are not statistically significantly different, at *p* < 0.05.

**Table 3 molecules-30-04569-t003:** Characteristics of sage (*Salvia officinalis* L.) according to the manufacturer’s declaration.

Sage Code	Company Code	Country of Origin	Form	Part of the Plant	Type of Cultivation
1LC	1	No information	Loose	Cut sage	Conventional
2LC	2	No information	Loose	Leaf	Conventional
3LC	3	No information	Loose	Leaf	Conventional
4BC	4	No information	Bag	Leaf	Conventional
5BC	5	No information	Bag	Leaf	Conventional
6LE	6	Bosnia and Herzegovina	Loose	Leaf	Ecological
7LC	7	Poland	Loose	Leaf	Conventional
8LE	8	Poland	Loose	Herb	Ecological
9LC	9	No information	Loose	Leaf	Conventional
10BE	10	Poland	Bag	Leaf	Ecological
11BC	11	No information	Bag	Leaf	Conventional
12BC	12	No information	Bag	Herb	Conventional
13LE	10	Poland	Loose	Leaf	Ecological

## Data Availability

The data used to support the findings of this study are available from the corresponding author upon request.

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
