# Peer review of "Content of Carnosic Acid, Carnosol, Rosmarinic Acid, and Proximate Composition in an Assortment of Dried Sage (*Salvia officinalis* L.)"

_molecules, 2025, doi:10.3390/molecules30234569_

Round 1
Reviewer 1 Report
Comments and Suggestions for Authors
1. (line 84-86) However, there is no data on the content of key polyphenolic compounds in dried sage, especially carnosic acid, carnosol and rosmarinic acid... (end of quote)
There is data, in more different references [REF1-REF4].
[REF 1] Plants 25;11(5):625.
[REF 2] Industrial Crops and Products 49 (2013)
[REF 3] Plant Physiol. Biochem. 2010;48:813–821.
[REF 4] J. Plant Nutr. Soil Sci. 2011;174:504–514.
And these references already show a wide range of different values for these and other polyphenolic compounds (unconsidered in this publication). So, I would suggest to rephrase your quoted sentence from above (lines 84-86).
2. Your analysis is conducted on four samples from Poland, one from Bosnia and Herzegovina and eight of unknown origin. What does the unknown origin mean, exactly? You state the following:
(lines 189-190) The remaining sages could therefore have originated from different European regions with different climatic conditions. (end of quote)
How sure are you in that? I ask because Bosnia and Herzegovina is outside of European Union. So, can any sample be outside of Europe (e.g. northern Africa or Asia)?
In addition, Bosnia and Herzegovina is sample 6LE, while Polish samples are 7LC, 8LE, 10BE and 13LE. Now please compare 6LE with samples 7LC, 8LE, 10BE and 13LE visually on Figure 1. Do you see a difference between 6LE and Polish samples? That tells me something, but the issue is that other samples are of unknown origin...
3. I took some time to inspect Tables 1 & 2. I see potential ambiguities.
A) First of all, where is information about number of independent measurements (n) from which each value in the table is obtained as mean ± standard deviation? Did all samples have equal n?
B) How sure are you that all lowercase (superscript) letters (a,b...) are correctly assigned accross all values? For example, you state that regarding Flavonoids originating from Poland:
7LC 13,9 ±0,6 c; 10BE 11,4 ±1,5 cd; 13LE 9,9 ±0,6 e; How is that possible that values 13.9±0,6 and 11,4 ±1,5 are mutually insignificant (mutual difference 13,9 - 11,4 = 2,5); while at the same time 11,4 ±1,5 and 9,9 ±0,6 are significantly different (mutual difference 11,4 - 9,9 = 1,5). 1,5 difference is significant and 2,5 isn't.
The other example is 7,1 ±0,1 f g (what does non-lowercase "f" stand for here?) and 4,8 ±0,3. Are you sure that they are not significantly different?
C) Table 2. Fat values.
12BC 6.21 ±0.84 e. 4BC 6.63 ±0.06 df. So, samples 12BC and 4BC are significantly different although their difference is only 0.42... However, regarding other samples: 7LC 8.73±0.40 de, 8LE 8.44 ±0.50 de, 9LC 9.42 ±0.07 de, 10BE 9.87 ±0.37 ce. We see here that these four samples (7LC - 10BE) are labeled as not significantly different to sample 12BC (same label "e") although the differences of their values with 12BC are 2.52 (as 8.73 - 6.21), 2.23, 3.21, 3.66, respectively. How can 0.42 be significant difference, while all four other differences (spanning the range 2.23-3.66) are not significant? How is that possible?
Reviewer 2 Report
Comments and Suggestions for Authors
Introduction
- There are some repetitive sentences in the introduction. Please review them to improve the flow.
- Emphasize the aim of this study clearly.
Results and Discussion
- In general, specify mg/g of what each time you report a value.
- It is better to improve discussion in relation to the aim of this study and compare your results with those reported in the literature.
Paragraph 2.1
Lines 105: Specify the content of carnosic acid. Is it expressed as mg/g of dry extract?
Lines 105–106: Report the value of carnosic acid in the sample, similar to how you reported the value of rosmarinic acid in line 149.
Lines 130–132: Carnosol is a derivative of carnosic acid. Explain why, in sample 12BC, the content of carnosic acid is low while carnosol is present, whereas in sample 9LC there is more carnosic acid but no detectable carnosol. Clarify this point.
Line 158: Define the extract that was evaluated in your previous study.
Line 176: Be more specific with the term “Others.”
Line 209: It is better to report the range rather than the average.
Line 212: Report the value for each sample individually.
Paragraph 2.2
In general, it is not necessary to report the standard error in the text; reporting only the mean value is sufficient.
Line 299: Replace “the fat content and ash content was generally” with “the fat content and ash content were generally.”
Materials and Methods
Paragraph 3.1
Please describe the materials used in this study more clearly. For example, the difference between sample 2LC and 3LC, and between 4BC and 5BC, is unclear from the table.
Paragraph 3.3
- Specify which samples were extracted with methanol and which with hydroalcoholic solution.
- Why was methanol chosen as the extraction solvent? And did you choose this specific extraction method?
- Did you perform an experimental design for the extraction?
- Were the extracts used directly after extraction, without solvent evaporation?
Paragraph 3.4
Specify the concentrations used for total polyphenols and flavonoids determination.
Paragraph 3.5
- Did you use standards for determining carnosic acid, carnosol, and rosmarinic acid?
- Would it be possible to provide the chromatograms for your samples?

Reviewer 3 Report
Comments and Suggestions for Authors
The manuscript presents valuable data regarding the variation of bioactive compounds, basic nutrients contents and phenolic acids in commercial dried sage products. However, several major and minor aspects require clarification or improvement before the paper quality.
Therefore, I suggest the following:
Major Revision:
- In the article it is mentioned the analysis of “minerals,” yet the study only determined total ash content, without individual elemental quantification (e.g., Ca, K, Mg, Fe). It would be more accurate to replace “minerals” with “ash content” or “total mineral residue” to avoid misleading interpretation.
- I suggest specifying the limitations of the used extraction method (the reflux extraction at 85 °C, as such conditions may lead to partial oxidative degradation of carnosic acid and alter the native diterpenic balance) and indicating whether the extracts were analysed immediately after preparation, since prolonged exposure or storage could promote oxidation or degradation of the target compounds.
- The description of total polyphenol (TPC) and total flavonoid (TFC) determinations is ambiguous and should be revised for clarity and reproducibility. It is recommended that these procedures be presented in two separate subsections, with the original specifications of the standards clearly indicated (including the type of standard compounds, calibration ranges, and regression parameters). Also specify the number of replicates for each analysed sample.
- Statistical analysis: The manuscript does not specify the number of analytical replicates (n). While ANOVA is mentioned, the statistical discussion is minimal, and correlation analyses between parameters (e.g., CA–C–RA, TPC–TFC) are missing, which would strengthen data interpretation.
Minor corrections:
(line 281) I suggest using “Bioactive compounds” instead of “Bioactive ingredients,” as it better reflects standard scientific terminology.
(line 325): The name of the reagent is misspelled. Please correct “Folin-Cocaulteu” to “Folin–Ciocalteu”.
(line 368): Please correct “ANAVA” to “ANOVA”
(line 547): Please verify the correctness of the AOAC reference.
Comments on the Quality of English LanguageThe manuscript is understandable but requires moderate English editing for style, terminology consistency, and correction of typographical errors. Minor grammatical adjustments and uniform use of decimal points and units are also recommended. Overall, the paper is readable, but professional editing is advised before publication.
Round 2
Reviewer 1 Report
Comments and Suggestions for Authors
The authors significantly improved the manuscript by following my recommendations, so I am pleased to confirm that the manuscript is ready for publication.
Just one last detail (line 231): "The remaining sages could therefore have originated from different regions of the word"... - I think it should be "world" instead of "word".
Reviewer 2 Report
Comments and Suggestions for Authors
The manuscript has been improved.